# Helping Humans Become Better Teachers for Robots with Augmented Reality

Andre Cleaver
andre.cleaver@tufts.edu
Tufts University
Medford, Massachusetts, USA

Jivko Sinapov
jivko.sinapov@tufts.edu
Tufts University
Medford, Massachusetts, USA

## ABSTRACT

We demonstrate *TRAinAR*, an augmented reality (AR)-based tool that is designed to improve sim2real reinforcement learning (RL) for robots. *TRAinAR* aims to enable users to train a robot by quickly prototyping complex environments in a virtual training environment with constraints to match the real-world. *TRAinAR* also allows a user to visualize the robot's training data in context of the environment which potentially can provide insights into ways to improve the robot's training process. In this paper, we propose a human-participant study to evaluate *TRAinAR* as a valuable training tool. The proposed user study will help humans better identify ways to teach and improve a robot's learning process. In a technical demonstration, our application enabled a robotic arm manipulator to learn how to navigate its end-effector toward a goal object while implicitly learning to avoid obstacles.

## KEYWORDS

Augmented-Reality, Data-Visualization, Reinforcement-Learning

**ACM Reference Format:**
Andre Cleaver and Jivko Sinapov. 2023. Helping Humans Become Better Teachers for Robots with Augmented Reality. In *Proceedings of Companion of the 2023 ACM/IEEE International Conference on Human-Robot Interaction (HRI '23 Companion).* ACM, Stockholm, SE, 4 pages. https://doi.org/XXXXXXX.XXXXXXX

## 1 INTRODUCTION

Teaching by demonstration is a common approach to train intelligent robots for personalized tasks, where a teacher will manually control (teleoperation, kinesthetic programming) a robot to complete a task in order to speed up or improve the robot's learning process. Training a robot often takes place in a controlled setting such as a research lab by roboticists; however, trained robots are expected to fail when deployed to new environments (i.e., homes, public spaces) due to changes in perceptions that differ from the initial trained setting and thus will require retraining.

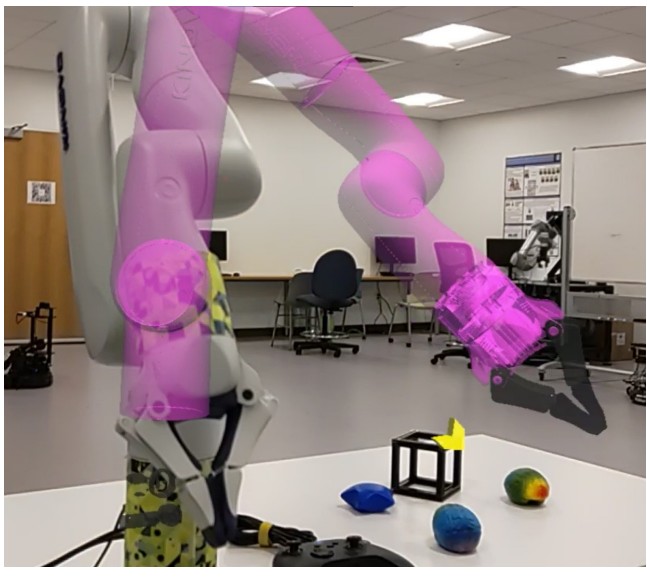

**Figure 1: Kinova Gen3 Lite robot with augmented digital twin visualized conveying motion intent (purple arm) towards target goal pose (yellow arrow).**

Users unfamiliar with robots are challenged when needing to teach a robot a new personalized task because of the lack of communication of what has been learned resulting in the user making poor mental models of the robot; therefore, it is crucial that a teacher receives adequate feedback from a robot in the teaching process. Augmented reality (AR) technology has shown to be an alternative medium in conveying complex robotic information through simplified visualizations [3–5, 16], and therefore, we hypothesize that visualizing the robotic learning process, which is often a "black box" to many individuals, may provide useful information for a user to train a robot effectively. Specifically, we ask the questions, "Can humans become better teachers to robots with AR technology?" and, "What information visualized do users benefit the most when training and teaching a robot?"

To address these questions, we propose a human-participant study where users are instructed to improve a robot's task with the help of an AR device. We developed **TRAinAR** (Teaching Robots Actions in Augmented Reality) which aims to enable a user to 1) quickly prototype a 3D virtual training environment with constraints that reflect the real-world, 2) visualize the robot's training data and learning parameters, and 3) animate a robot's future actions before execution. Results from the proposed user study will

determine *TRAinAR* as a viable tool for robotic sim2real training and provide insights into visualized robot data for improving the training process. In a technical demonstration, our application enabled a robot to learn how to navigate its gripper toward a target object while implicitly avoiding obstacles.

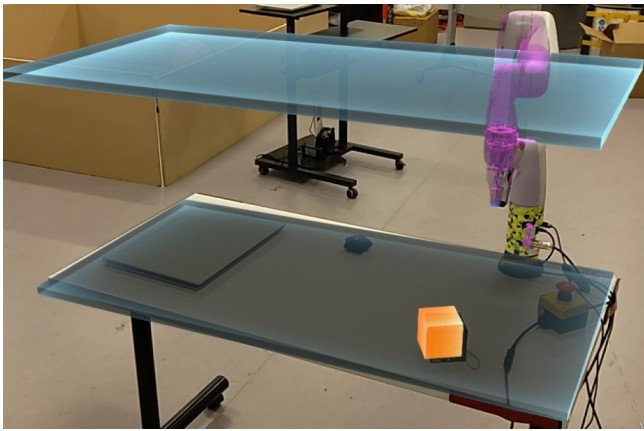

**Figure 2: Kinova Gen3 Lite robot with virtual boundary planes (ground and ceiling) that restricts both the robots movement as well as the exploration space.**

## 2 RELATED WORK

Explainable robotics and AI is an emerging field that aims to address the "black box" issue regarding decision making by advanced robotics or artificial intelligent (AI) systems that control a robot [11]. Reinforcement learning (RL) and machine learning algorithms also fall under this category where the decision pipelines are not transparent making it difficult for humans to put their trust with a robot [20]. Additionally, the lack of an explanation behind a robot's decisions can result in mismatched mental models between a user and the robot ultimately leading to undesired outcomes [7, 8, 22]. As a result, studies have looked into robotic systems that convey their decision process to humans. [6] evaluated robotic systems in an RL setting through a goal-driven approach with robot decisions explained in terms of probabilities of success for easy interpretation. [1] demonstrated virtual robots that were perceived as more lively and human-like when explaining gaming strategies out loud. [24] proposed an experimental framework combining AR with explainable AI with a virtual robot. The AR system visualized the perceptual beliefs of the robot as the user interacts with virtual objects that were part of a daily activity.

More notable works explored training a virtual agent in a virtual environment using neural networks and later transferring the knowledge to a real robot. [9] and [21] both trained robotic systems using humans demonstrating grasping techniques in virtual reality that later transferred to a physical robot. [25] similarly trained a robotic system across several types of object manipulation tasks with virtual reality teleoperation. [14] demonstrated constrained learning applied to a robotic end effector with augmented reality to improve humans teaching robots various skills. Inspired by the *ARC-LfD* project by [15], we aim to create *TRAinAR* as a tool to quickly prototype complex environments in virtual environments with a similar form of augmented constraints. In our proposed human-participant study, participants will train a robotic arm manipulator given the capabilities to modify the virtual training environment as well as access to the robot's existing knowledge of the task.

Augmented Reality technology (AR) has been leveraged by the robotics community as a useful tool to communicate robotic information such as robotic trajectory planning [18], robot motion intent [23?], safety zones [10], and state [2]. [13] demonstrated an AR-based system that only provides a user visual feedback regarding the desired target goal position for robotic arm manipulators on a ground industrial mobile robot. The AR system enabled users the opportunity to correct or re-plan a trajectory after visualizing the planned trajectory in context of the workspace. Following a similar approach, we aim to visualize training data using AR to determine if users gain insights into ways to improve the robot's training process.

## 3 METHODOLOGY

To determine *TRAinAR* as a viable training tool, we propose a between human-participant study. Participants are assigned to either the test group or control group. The difference between the groups is how the information is visualized: on a standard computer monitor or through an augmented reality device. The control group will only be interacting with a laptop computer, and the test group will interact through the augmented reality (AR) device.

For each participant, the entire session is split into three parts: a pre-questionnaire with tutorial, robot training, and a post questionnaire. The tutorial will focus on how to use *TRAinAR* along with a tutorial on how to control the robotic arm manipulator via game controller. Participants in the test group will be fitted with the Hololens2 and shown how to interact with the virtual environment through hand gestures. Participants in the control group will only use a standard laptop computer with a mouse. Both groups will proceed to complete the following tasks:

### 3.1 Task Descriptions

*3.1.1 Modeling the Environment.* In this task, the participant will modify a virtual training environment to constrain the robot to stay in its working zone and avoid collisions with other objects. The participant will set the constraints for the robot by repositioning the virtual objects that correspond to its real-world counterpart. For example, a participant will reposition a bottom boundary plane that corresponds to the physical table that was purposely misplaced. The participant in turn is expected to reposition the virtual plane to match the height of the physical table or superimpose the virtual plane over the physical table. Participants wearing the AR device will perform a pinch gesture to manually reposition the virtual plane, and the participants using the computer will use a mouse to click and drag the virtual plane to where they believe the plane should be placed. The changes to the virtual environment will be recorded and then recreated by the PI for later training.

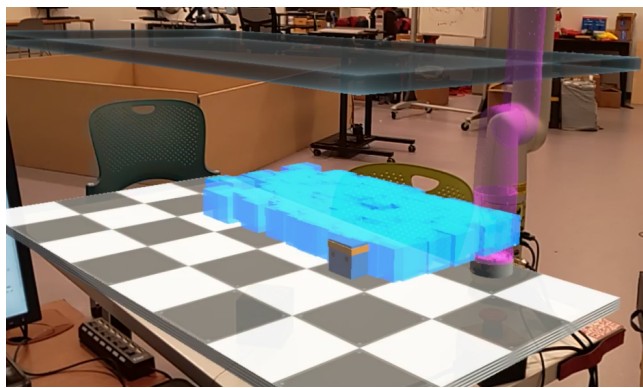

**Figure 3: The user visualizes the training data history over the real-world environment. The agent here is represented as a blue cube which spawns in random locations for each training episode. Users with visualized training data can gain insights into the agent's learning performance.**

*3.1.2 Teaching by Demonstration.* In this task, we will instruct the participant that the task the robot will learn is navigating its end-effector (gripper tip) to a target object which can be seen highlighted in orange in Figure 2.The participant will use a gaming controller to move the robot's arm to the object's location. The actions by the participant will be recorded, and the process of demonstration repeats for a set period of time.

*3.1.3 Knowledge From the Robot's Training History.* We include this task to investigate what a user can determine from the robot's training history data and how they would improve the robot's training process. For example, Figure 3 shows a history of an agent's spawned locations visualized in context after a training session. In a questionnaire, participants report sections where the robot has already explored and where the robot should explore next to expand its knowledge to improve its performance.

## 3.2 Hypotheses

**H1**: Users with AR will train the robot more confidently compared to users without AR.

**H2**: Users with AR feedback will require shorter training times to train the robot.

**H3**: Robots that have been train by users with AR feedback will have greater performance compared to robots that have been trained by users without AR feedback.

**H4**: Users that train the robot with AR will report lower cognitive load compared to users that train the robot without AR.

## 3.3 Hardware & Software

We demonstrate *TRAinAR* using a 6 degress of freedom *Kinova Gen3 Lite* robot controlled with *Robot Operating System* (ROS) [17] running on Ubuntu 18.04. Python and C++ scripts filtered the robotic data before delivery to the AR device for visualization. *TRAinAR*

was developed with *Unity*[1] and deployed onto a (*Hololens2*) which rendered the robotic data over the physical environment.

*Vuforia*[2] tracked the robot's position and orientation through a cylindrical target-image fixed to the robot (see Figure 1). C# scripts control data exchange between the robot and AR device that occurred over a shared Wi-fi network using *ROS Sharp*[3]. We utilized *Unity ML-Agents Toolkit* [12] to train a virtual agent that represented the real robot. We leveraged Microsoft's Mixed Reality Toolkit 2 (MRTK2) to handle user hand gesture tracking when manipulating with the virtual objects. Additional C# scripts logged the training history data such as the virtual agent's position throughout the training process.

## 3.4 The Virtual Training Environment

For simplicity, we aim to have participants train the robot to perform a classic move-to-target task; specifically, have the robot navigate its end-effector to a target object while avoiding obstacles. Unity's ML-agents toolkit by default uses proximal policy optimization to map the agent's observations to the best actions in a given state. In our case, our agent will learn to navigate from an initial starting position to a target object in 3D space. From Figure 2, the virtual planes are boundary constraints to limit the exploration space. A digital twin of the robotic arm is added to the virtual environment along with virtual objects to represent any additional obstacles that will be randomly placed in the real-world for the participants to account for. All virtual objects have rigid-body and collider components which enable users to interact with them. The training agent which represents the robotic end-effector has a unique script that terminates a training episode once it makes collides with any object. Positive and negative rewards are assigned to the target object and obstacles respectively. We train the agent in continuous action space with 3 actions that correspond to x-y-z velocity inputs in Cartesian coordinate space. The agent's observation space includes the x-y-z positions of both the target object and the robot end-effector agent.

For teaching by demonstration, Unity's ML-Agents Toolkit has a *Demonstration Recorder* script that captures the user's actions for imitation learning. During the training phase of the study, participants will use a gaming controller to send actions to control the robot's end-effector. These actions will then be incorporated into the robot's learning algorithm to hopefully speed up the learning process.

## 3.5 Measures & Analysis

To answer our questions, we plan to gather a combination of objective and subjective measures. Both the Hololens2, laptop, and robot connect with ROS; therefore, we plan to record timestamped data for environment changes and user actions. We will consider overall demonstration times for both conditions, expecting shorter times for the test group (**H2**). The total training time in simulation for the agent is also expected to be shorter for the test group. The average reward collected in training is expected to be greater in the test group (**H3**). After completion of the tasks, participants will be

---

[1]https://unity.com/

[2]https://developer.vuforia.com/

[3]https://github.com/siemens/ros-sharp

administered a questionnaire to gauge how effective they believe they were in training the robot. User confidence will be recorded in a Likert scale which we anticipate greater ratings compared to the control group (**H1**). Measuring user cognitive load will be conducted with the NASA Task Load Index, expecting lower ratings for the test group (**H4**). Open-ended questions will prompt participants to describe helpful visuals.

## 4 DISCUSSION

We present a proposed human-participant study to evaluate *TRAinAR* as a viable tool for improving the sim2real training process for Reinforcement Learning and robots. *TRAinAR* aims to enable users to learn more about robot's personalized task by visualizing the training environment and data history to shed light on any unexpected robot behaviors. As a result, users can modify a virtual training environment to improve the training process. In a technical demonstration [4], we show that a robot was able to learn to move towards a target object after tailoring a virtual training environment described in section 3.4.

*TRAinAR* in its current stage does not allow a user to provide real-time input to the robot; training is still happening without any explicit human help during the process. Users only have manual control over constraint placement in the virtual environment prior to training. We are currently developing *TRAinAR* for preference-based learning [19], where users are prompted to rank actions or trajectories generated by the robot which can also be visualized in AR. A successful outcome for *TRAinAR* will allow further investigation for more complex robot tasks and environments.

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

---

[4]Video Demonstration