# OpenReview forum: "Helping Humans Become Better Teachers for Robots with Augmented Reality"
_humanrobotinteraction.org/HRI/2023/Workshop/VAM-HRI — VAM-HRI 2023 Oral_

### Official Review · Program_Chairs · 2023-02-01
**Accept**

**Rating:** 7
**Confidence:** 5

**Review:**

Review 1

This paper presents TRAinAR, an AR tool for constructing virtual training environments to aid in sim2real RL for robots.

Overall, this paper investigates an interesting research direction. The most interesting component is visualizing the robot's training history during the learning process, although there are some questions regarding how it is implemented and used (detailed below). While the paper mentions investigating having a human provide feedback during the learning process will be for future work, I recommend they investigate this deeply since it will make the research much more impactful. Overall I recommend this paper be accepted.

Comment/Questions:

- Why does there need to be a virtual plane at the top of the scene to restrict the robot's exploration? The plane aligned with the table makes sense to prevent collisions, but the upper one is not clear.

- The methodology states that the difference between the test group and the control group will be how information is visualized, but there is also a difference in how users interact with the scene (game controller or pinch gestures), which should be emphasized and potentially investigated separately.

- What is the reason for having the user teach the robot via game controller for demonstrations rather than pinch-and-move virtual end effectors (like in citation [1] included below)?

- The agent's observation only includes the xyz of target object and robot end effector, but shouldn't it also include information about obstacles to enable it to avoid the objects? Or is the training environment the same as the test environment (i.e: obstacle location never changes)?

- how will collected data from users be used for imitation learning specifically? (e.g: will behavioral cloning happen first before RL, or will PPO loss and imitation learning loss be combined together?). how many demonstrations will users supply?

- The training information visualized in Figure 3 is a bit confusing and hard to understand, how does showing the robot's start locations help give insight into the agent's learning performance?

- footnote [4] seems like it should give a link to a video demonstration?

- In the "Knowledge from the robot's training history" section, it is not clear why having the human know where the robot's spawned locations would help improve the human's ability to teach the robot or have a better understanding of the robot's mental model/policy. Does the human determine where the robot starts?

- If the height of the table is known, why not measure how far off the table height error between each interface as a metric?

- Overall, there is very little information about the TRAinAR interface itself, with most details being included in Task Descriptions. I'd recommend making a separate section describing the system/tool details of the interface.


Minor issues:
- Reference [1] seems incorrect in bibliography
- In related work, one of the citations appears as a [?]
- Them agent's observation space includes -> [The] agent's observation space includes...

[1] Rosen, Eric, et al. "Communicating and controlling robot arm motion intent through mixed-reality head-mounted displays." The International Journal of Robotics Research 38.12-13 (2019): 1513-1526.


---------------------------------


Review 2

This paper examines how AR displays can enhance the training of robot manipulators. This planned work will utilize a digital twin to train a robot arm to move objects on a table. The experiment trials will look at three aspects of the training process: 1.) workcell constraints; 2.) training demonstrations; and 3.) training completeness/history. This work seems highly relevant to the VAM-HRI field and seems perfectly suitable for this workshop. This line of research seems highly useful for the HRI community as a whole, but my main caution for the authors is to: 1.) expand the related work to include additional VAM-HRI papers that use methods similar to that presented in the paper; and 2.) explicitly explain how the implemented system differentiates from prior work and/or adds a novel contribution to the VAM-HRI subfield of learning from demonstration via AR technology. Regardless I recommend this paper for acceptance.

Questions and Comments:

- The intro/motivation is strong and presents a solid argument as to why AR is poised to enhance robot learning from demonstration.

- Missing citation in the related work section.

- Some additional citations to consider adding include (https://vamhri.com/):

     o	Deep Imitation Learning for Complex Manipulation Tasks from Virtual Reality Teleoperation by Zhang, T., McCarthy, Z., Jow, O., Lee, D., Chen, X., Goldberg, K., & Abbeel, P.. ICRA (2017)

     o	Teaching a Robot to Grasp Real Fish by Imitation Learning from a Human Supervisor in Virtual Reality by Dyrstad, J. S., Øye, E. R., Stahl, A., & Mathiassen, J. R.. IROS (2018)

     o	Robot Learning from Human Demonstration in Virtual Reality by Stramandinoli, F., Lore, K.G., Peters, J.R., O’Neill, P.C., Nair, B.M., Varma, R., Ryde, J.C., Miller, J.T. and Reddy, K.K.. VAM-HRI (2018)

     o	Augmented Reality Interface for Constrained Learning from Demonstration by Luebbers, M. B., Brooks, C., Kim, M. J., Szafir, D., & Hayes, B.. VAM-HRI (2019)

     o	Arc-lfd: Using augmented reality for interactive long-term robot skill maintenance via constrained learning from demonstration. MB Luebbers

- I find the user placements of constraints in the modeling the environment task to be interesting and I am interested in knowing what other constraints (and methods of defining) can enhance these types of systems in more complex work environments.

- I think more information on the inputs to both the baseline system and the AR system would strengthen the paper (I am left wondering which components are controlled via motion controls/mouse+keyboard and game controller).

- How does the baseline interface work in the evaluation of training history? AR users can move around the workspace at will, how does the user manipulate the viewport of the monitor?

- “The training agent which represents the robotic end-effector has a unique script that terminates a training episode once it makes collides with any object” Does this include the objects the robot is to grasp and not just obstacles? If the robot does not demonstrate actually grasping the object I wonder how users will be able to evaluate “how effective they believe they were in training the robot”

---

### Decision · Program_Chairs · 2023-03-02

Accept (Oral)